# *S*-LIGHT: Synthetic Dataset for the Separation of Diffuse and Specular Reflection Images

**DOI:** 10.3390/s24072286

**Published:** 2024-04-03

**Authors:** Sangho Jo, Ohtae Jang, Chaitali Bhattacharyya, Minjun Kim, Taeseok Lee, Yewon Jang, Haekang Song, Hyukmin Kwon, Saebyeol Do, Sungho Kim

**Affiliations:** Advanced Visual Intelligence Lab (AVILAB), Yeungnam University, Gyeongsan-si 38541, Republic of Korea; jwhtkdghj@gmail.com (S.J.); otaejang@postech.ac.kr (O.J.); bhchaitali@yu.ac.kr (C.B.); kmj17211@gmail.com (M.K.); xotjr1949@yu.ac.kr (T.L.); zzolimi49@yu.ac.kr (Y.J.); songsunriver@gmail.com (H.S.); pos07039@yu.ac.kr (H.K.); lloeyvyu@yu.ac.kr (S.D.)

**Keywords:** single image based deep learning model, specular highlight removal, reflection removal, synthetic dataset, multi-scale normalized cross correlation (MS-NCC)

## Abstract

Several studies in computer vision have examined specular removal, which is crucial for object detection and recognition. This research has traditionally been divided into two tasks: specular highlight removal, which focuses on removing specular highlights on object surfaces, and reflection removal, which deals with specular reflections occurring on glass surfaces. In reality, however, both types of specular effects often coexist, making it a fundamental challenge that has not been adequately addressed. Recognizing the necessity of integrating specular components handled in both tasks, we constructed a specular-light (*S*-Light) DB for training single-image-based deep learning models. Moreover, considering the absence of benchmark datasets for quantitative evaluation, the multi-scale normalized cross correlation (MS-NCC) metric, which considers the correlation between specular and diffuse components, was introduced to assess the learning outcomes.

## 1. Introduction

Research in computer vision has utilized various types of image data, such as electric optical (EO), infrared (IR), and light detection and ranging (Lidar). Among these, EO images, which are within the visible spectrum perceivable by humans (380 nm to 780 nm), have been studied extensively. EO images have been used for research object detection, classification, image registration, 3D transformation, and other areas, due to their unique characteristics and patterns. The advent of deep learning has prompted significant growth in EO image-related research. On the other hand, deep learning models based solely on single EO images are not robust enough to perform well in various environments. Recent deep-learning research has addressed this issue by developing more robust models that combine sensors from EO and other modalities [1,2].

One of the causes of the drawbacks associated with EO images is the presence of specular reflection components. The term “specular reflection component” refers to the portion of light energy that impinges on the surface of an object and is reflected without change, which can degrade the performance of object classification, detection, and segmentation in computer vision. Efforts to address and eliminate these reflection components have been ongoing in computer vision research since its inception, and research to solve this problem continues to the present day. Specular reflection components vary in their characteristics depending on the material, which has led to research in two main categories: reflection removal and specular highlight removal.

**Specular highlight removal:** The problem in specular highlight removal can be described using Equation (Equation 1). Figure 1 provides an example image representing the diffuse and specular components occurring on the surface of an object. In Equation (Equation 1), the diffuse component is denoted as $I_{d}$, the specular component as $I_{s}$, and the combined intensity as *I*.
(1)$$\begin{array}{c} I=I_{d}+I_{s} \end{array}$$

Here, the diffuse component refers to the light rays that penetrate the surface of an object, undergo multiple reflections and refractions, and are then re-emitted. On the other hand, specular refers to the portion of incoming light rays that are reflected directly, and its intensity varies depending on the roughness of the surface of an object.

**Reflection removal:** Unlike the problem tackled by specular highlight removal, the primary goal of the reflection removal task is to eliminate reflections occurring in situations with a medium, such as glass, between the object and the viewer and to generate an image of the object that has passed through the glass. In Figure 2, transmission and reflection represent $I_{t}$ and $I_{r}$ in Equation (Equation 2).
(2)$$\begin{array}{c} I=I_{t}+I_{r} \end{array}$$

Each component undergoes transformations depending on the properties of the glass. The variable $I_{t}$ is in a state where it has experienced pixel shifts and color alterations due to the refraction and absorption effects of the glass, whereas $I_{r}$ is affected by reflection and refraction effects in Equation (Equation 2). Taking these physical characteristics into consideration, most techniques model $I_{t}$ and $I_{r}$ in Equation (Equation 2) as follows.
(3)$$\begin{array}{c} I_{t}=\alpha I_{T} \end{array}$$
(4)$$\begin{array}{c} I_{r}=\beta \left(I_{R}*k\right) \end{array}$$

In Equation (Equation 3), $I_{T}$ represents the image before passing through the glass, α denotes the weighting coefficient, and in Equation (Equation 4), β also represents the weighting coefficient. The image $I_{R}$ before being reflected by the glass undergoes convolution with the potential degradation kernel *k*.

The reflection removal task focuses on removing images reflected on glass surfaces to produce transmission images. In contrast, specular highlight removal aims to eliminate the specular component images on inhomogeneous object surfaces to generate the remaining image.

Research has been conducted using single and multiple images to address these two categories. In the early stages of research, the problem of removing specular reflections from a single image was considered an ill-posed problem, leading to studies that relied heavily on handcrafted priors. Using a single image allows easy data collection and high computational efficiency, but it is not robust under various environmental conditions. Researchers have also explored the use of multiple images to overcome the performance limitations of single images. These approaches include methods [3,4,5,6] based on multi-view images that leverage the viewpoint-dependent nature of the specular component. The approaches also include methods [7,8,9,10,11] based on multiple polarized filter images that exploit polarization when light reflects from object surfaces and some methods [12,13] based on flash and non-flash image pairs. Although research based on multiple images offered better performance and robustness than single-image techniques, there are practical limitations because of the challenges of acquiring multiple images in real-world settings and the high computational demands.

Recent advances in deep learning techniques have led to a significant breakthrough in removing specular highlights and reflection components from single images. Despite these advances, research on these two tasks is still being conducted independently, and they are not robust against each other’s tasks, which limits their practicality. Figure 3 displays images containing specular reflection components frequently encountered in everyday life.

In the real world, specular highlight and reflection do not occur selectively. Hence, we advocate for the need for research into unified specular reflection removal that tackles both tasks. However, there are major challenges in driving unified reflection removal research, and to the best of our knowledge, there is no relevant research to date. Acquiring diffuse and specular reflection images in the uncontrollable conditions of the real world is nearly impossible. This issue not only prevents the use of real images as a training dataset but also precludes the possibility of conducting quantitative evaluations.

To alleviate this issue, this paper proposes a training dataset for single image-based deep learning models that considers recent developments in deep learning techniques and practical considerations. The paper also discusses which category of deep learning models is suitable for training and demonstrates their effectiveness in real-world images. The contributions are as follows:First, a synthetic dataset was constructed for training a single-image-based deep learning model that can consider both the specular components reflected on the surface and those reflected on the glass, a consideration not present in traditional specular highlight and reflection removal.Second, this paper proposes a performance metric that considers the relationship between the diffuse component and each specular component to measure the separation level quantitatively. Additionally, it offers the advantage of being able to visualize the results.

This paper is structured as follows. Section 2 covers non-learning and deep-learning methods for specular highlight removal and reflection removal tasks and the datasets used for training deep-learning models. Section 3 addresses the dataset that considers integrated specular components and the performance metric that takes the correlation between the diffuse component and the integrated specular component into account. Section 4 presents experimental comparisons of the proposed dataset and existing datasets regarding the learning performance and generalization, along with the results of applying the methods to real-world images. Finally, Section 5 reports the conclusion and outlines future research directions.

The energy of wavelengths in the visible spectrum undergoes numerous interactions before reaching the camera and the human eye. Modeling these phenomena remains a critical research topic in computer graphics and computer vision. Section 2 overviews how previous studies tackled these two categories. The section is divided into two parts, with the first and second parts discussing recent trends and the limitations of existing methods, respectively.

## 2. Related Work

This section comprises three subsections, each covering the approach taken for the respective tasks. First, Section 2.1 describes non-deep learning (Non-DL) techniques; second, Section 2.2 discusses deep learning (DL) techniques; and finally, in order to prepare a dataset for the deep learning models, Section 2.3 details the methods for obtaining training data.

### 2.1. Non-Deep Learning (Non-DL)

*Specular Highlight Removal:* Removing specular highlights has been a subject of research for a long time. Shafer [14] proposed a dichromatic reflection model to examine the components present on the surface of an object. This model was formulated assuming the object was opaque and inhomogeneous. The dichromatic reflection model has been widely cited and has significantly influenced the field of computer graphics and subsequent research in specular highlight removal. Klinker et al. [15] extended the concepts of the research [14] by considering that the distribution of pixels on the plane follows a t-shaped color distribution. They proposed an algorithm based on this concept. Bajcsy et al. [16] suggested a technique that performs this task in a color domain consisting of lightness, saturation, and hue. On the other hand, these techniques require color segmentation techniques, which can be a significant drawback in complex textures.

Tan and Ikeuchi [17] aimed to overcome these issues by proposing an effective method that can remove specular highlights without the need for explicit color segmentation. They introduced the specular-to-diffuse mechanism and the concept of a specular-free image containing only the diffuse reflection component. Based on the observation that the specular-free image has the same geometric distribution as the original intensity image, they devised a method to estimate the diffuse reflection through the logarithmic differentiation of the original and specular-free images. This approach significantly influenced further research [18,19,20,21].

Subsequent research focused heavily on optimization techniques. Kim et al. [22] proposed a maximum a posteriori (MAP) optimization technique based on the observation that the dark channel provides pseudo-specular reflection results in general natural images. Fu et al. [23] used k-means clustering and various priors in an optimization technique, and Akashi and Okatani [24] proposed an optimization technique based on sparse non-negative Matrix factorization (NMF).

*Reflection Removal:* The work of Levin et al. [25] is the first proposed method for separating mixed images with reflections using a single image. It introduced an optimization algorithm that minimizes the total amount of edges and corners in the separated images through a prior that they should be minimal. On the other hand, this algorithm was not robust in handling complex pattern images. Removing reflections using a single image is highly ill-posed.

Various studies have been conducted based on different priors and assumptions to make it more tractable. Levin and Weiss [26] proposed a method that utilizes gradient sparsity priors after users specify a few labels on the image. Li and Brown [27] used the fact that the camera focus is adjusted for a transmission image, proposing an optimization technique that forces the gradient components of the transmission image to have a long-tailed distribution and those of the reflection image to have a short-tailed distribution. Shih et al. [28] modeled the ghost effect using a double-impulse convolution kernel based on the characteristics of ghosting cues. They proposed an algorithm using the Gaussian mixture model (GMM) to separate the layers. Wan et al. [29] introduced a multi-scale depth of field (DoF) computation method and separated background and reflection using edge pixel classification based on the computed DoF map. Wan et al. [30] used content priors and gradient priors to automatically detect regions with reflections and those without reflections. They proposed an integrated optimization framework for content restoration, background-reflection separation, and missing content restoration.

### 2.2. Deep Learning (DL)

*Specular Highlight Removal:* Traditional non-DL methods removed reflection components using handcrafted priors. Nevertheless, one of the drawbacks of handcrafted priors is their lack of robustness in various environments. Recent research has shifted towards data-driven learning approaches to address this issue.

Funke et al. [31] proposed a GAN-based network to remove specular highlights from single endoscopic images. Lin et al. [32] introduced a new learning method: a fully convolutional neural network (CNN) for generating the diffuse component. Unlike traditional GAN methods, this approach used a multi-class classifier in the discriminator instead of a binary classifier to find more constrained features. Muhammad et al. [33] introduced Spec-Net and Spec-CGAN to remove high-intensity specular highlights in low-saturation images, particularly faces.

The DL techniques mentioned above can still result in color distortion in areas without specular components. Therefore, research has focused on detecting specular highlight regions and removing only the specular components in these areas. Fu et al. [34] proposed a generalized image formation model with region detection and a multi-task network based on it. Wu et al. [35] introduced a GAN network that models the mapping relationship between the two component image areas using an attention mechanism. Hu et al. [36] considered that specular highlight components have peculiar characteristics in the luminance channel and proposed a Mask-Guided Cycle-GAN. Wu et al. [37] presented an end-to-end deep learning framework consisting of a highlight detection network and a Unet-Transformer network for highlight removal. On the other hand, mask-guided techniques have limitations in removing large-area specular components.

*Reflection Removal:* Due to the success of deep convolutional neural networks in computer vision tasks compared to non-DL methods, researchers have proposed new data-driven methods for generating robust transmission image predictions for various reflection types. Fan et al. [38] introduced the first reflection removal deep learning model that utilizes linear methods to synthesize images with reflections for training and uses edge maps as auxiliary information for guidance. Zhang et al. [39] proposed an exclusion loss that minimizes the correlation between the gradient maps of the estimated transmission and reflection layers based on the observation that the edges of transmission and reflection images are unlikely to overlap. Yang et al. [40] introduced a deep neural network with a serial structure alternating between estimating reflection and transmission images. Li et al. [41] drew inspiration from iterative structure reduction and proposed a continuous network that iteratively refines the estimates of transmission and reflection images using an LSTM module. Zheng et al. [42] were inspired by the idea that the absorption effect can be represented numerically by the average of the refraction amplitude coefficient map. They proposed a two-step solution where they estimated the absorption effect from the intensity image and then used the intensity image and the absorption effect as inputs. Dong et al. [43] proposed a deep learning model to identify and remove strong reflection locations using multi-scale Laplacian features. Wei et al. [44] proposed method to enhance a baseline network with context encoding modules to use high-level context for clarity in reflective areas, and a novel loss function that leverages easier-to-collect misaligned real-world data. Li et al. [45] introduced a two-stage deep learning network that first estimates the reflection layer and then utilizes a Reflection-Aware Guidance module to improve the transmission layer prediction.

### 2.3. Training Datasets Construction

*Specular Highlight Removal:* As acquiring ideal specular and diffuse components in real-world settings is virtually impossible, constructing datasets has been carried out by constraining the shooting environment or applying constraints. Two datasets have been proposed for training deep learning models for specular highlight removal.

The first dataset, the specular highlight image quadruples (SHIQ) dataset [34], utilized the MIW dataset [46] acquired by capturing flashes from various directions, which was publicly available. They obtained reflection-free images from this dataset using existing multi-image-based reflection removal techniques [3]. They further selected high-quality images from the results. After unifying the specular regions to white, they constructed the dataset by cropping and producing specular region masks, resulting in intensity, diffuse, specular, and mask pairs. The second dataset, the PSD dataset [35], was constructed from real images using a physically based polarizing filter. They applied linear polarizing filters to the light source and circular polarizing filters to the camera, capturing images with minimal values in the filter regions. This process allowed them to obtain images with removed specular components on the object surface, forming the dataset.

*Reflection Removal:* Building a large-scale dataset containing reflection components is a challenging task. Therefore, various strategies have been used to produce training datasets, including generating mixed images by adding reflection and transmission images produced based on mathematical formulae. Alternatively, real-world reflection and transmission images have been acquired separately and combined through linear blending. Some approaches utilize real and synthetic images [38,47,48,49]. Datasets captured in various environments have been proposed to evaluate the network performance. Among them, the prominent dataset is $SIR^{2}$ [50]. $SIR^{2}$ is categorized into solid objects, postcards, and wild scenes. For solid objects and postcards, the images were captured from seven DoFs and three glass thicknesses in 20 scenes. Wild scenes comprised 100 scenes captured under various settings, including different glass thicknesses, camera settings, and uncontrolled illumination conditions.

Existing datasets for specular highlight removal each have their limitations as follows: The SHIQ dataset is built on the assumption that specular components are white, making it unable to remove specular components of various colors. The PSD dataset has low diversity due to its limited environments or objects and is composed of specular areas with narrow coverage, limiting the removal of specular areas of various sizes. Datasets for reflection removal are designed to eliminate reflection components, thus failing to consider specular components in the transmitted scene. For these reasons, this paper proposes the necessity of constructing a new form of dataset to eliminate integrated specular reflection for the first time.

## 3. Proposed Method

This section is structured to explain the methods for constructing the specular light (*S*-Light) dataset, designed to address the removal of specular components occurring in various scenarios, as well as to introduce the performance metric multi-scale normalized cross-correlation (MS-NCC) to measure the correlation between specular and diffuse components.

To address the limitations of existing datasets mentioned in the last paragraph of Section 2.3, we have constructed the database in two stages. Figure 4 illustrates an overall workflow for constructing our proposed *S*-Light DB. In the first step (step-1 in the figure), we collected 3D models that were rendered in Blender to deal with the specular highlight removal problem. Subsequently, in the next step (step-2 in the figure), we used methods [38,47] to handle the reflection removal issue.

### 3.1. Dataset Construction

As shown in Figure 5, the Case 1 dataset, $S{\text{-}\mathrm{Light}}_{R}$, consists of *I*, $I_{t}$, and $I_{r}$, whereas the Case 3 dataset, $S{\text{-}\mathrm{Light}}_{C}$, comprises *I*, $I_{td}$, and $I_{r}$ + $I_{ts}$. Figure 6 shows example images for each case. Each subscript in *S*-Light represents surface specular, reflection, and combined, respectively.

#### 3.1.1. Step 1: Construction of $S{\text{-}\mathrm{Light}}_{S}$ Dataset

The SHIQ and PSD datasets consider Case 2 scenarios. Existing datasets have the following issues. The SHIQ dataset contains various objects but suffers from specular highlight colors composed only of white, making it less robust to various lighting conditions. PSD, being captured in a controlled laboratory environment, often exhibits mostly black backgrounds or repetitive patterns, and it lacks diversity in the types of objects.

Therefore, 130 multi-object 3D models were collected from the open-source platform Blend Swap [51]. The collected 3D models encompass a variety of object types with different materials (such as metals and plastics), textures (smooth and rough), and shapes (curved surfaces and sharp edges). These models were used in Blender (v3.4), a free and open-source 3D computer graphics software, using the Path Tracing-based Cycles engine [51], which employs materials and lighting grounded in physical laws, ensuring predictable and realistic outcomes. Seven hundred and nineteen scenes were generated in rendering scenes where objects and backgrounds do not overlap. Each scene was rendered into intensity and diffuse images at a resolution of 1024×1224. For data augmentation, the images were cropped randomly to a 512×512 resolution or resized, resulting in a dataset of 3595 pairs.

Table 1 presents a quantitative comparison of the existing datasets, SHIQ, PSD, and the dataset developed in this study. The letters I, S, D, and M stand for intensity, specular, diffuse, and mask, respectively. “Clustering portion” indicates the proportion occupied by the largest cluster. Dimensionality reduction into two dimensions using t-SNE [52] and applying the DBSCAN clustering algorithm [53] was performed for each dataset. The t-SNE technique reduces dimensions in such a way that similar data points are clustered closely together, and DBSCAN clusters data points based on their density. Therefore, having many cluster groups with few points in each group suggests that the image dataset exhibits specific patterns and lacks diversity. Consequently, a higher “clustering portion” indicates that the dataset is more diverse and has a uniform density. Figure 7 provides examples of the results of applying the t-SNE and DBSCAN algorithms.

The distribution of the proportion of specular areas within the entire image for our constructed $S{\text{-}\mathrm{Light}}_{S}$, as well as for SHIQ and PSD, can be examined in Figure 8a. It demonstrates that, in contrast to $S{\text{-}\mathrm{Light}}_{S}$ and SHIQ, which feature regions (specular area) of various sizes, PSD is concentrated in distributions of narrower areas. Additionally, Figure 8 b–d directly show that, whereas $S{\text{-}\mathrm{Light}}_{S}$ and PSD contain specular components in various colors, SHIQ is limited to a single color (white) for its specular reflections. In summary, $S{\text{-}\mathrm{Light}}_{S}$ is a dataset that encompasses diverse image patterns, specular colors, and specular regions of various sizes, effectively addressing the limitations observed in SHIQ and PSD.

#### 3.1.2. Step 2: Construction of $S{\text{-}\mathrm{Light}}_{R}$ and $S{\text{-}\mathrm{Light}}_{C}$ Dataset

Previous research has proposed various methods to generate synthetic images and build the reflection-removal dataset. In this study, the techniques reported elsewhere [38,47] were applied to generate reflection images. Figure 4 shows the Step 2 process of constructing $S{\text{-}\mathrm{Light}}_{R}$ and $S{\text{-}\mathrm{Light}}_{C}$ as a block diagram. To build $S{\text{-}\mathrm{Light}}_{R}$ and $S{\text{-}\mathrm{Light}}_{C}$, the I and D from $S{\text{-}\mathrm{Light}}_{S}$, constructed in Step 1, were considered as $I_{T}$ according to Equation (Equation 3). For generating reflection images, the Pascal VOC dataset [54] was regarded as $I_{R}$ according to Equation (Equation 4). Additionally, for constructing the $S{\text{-}\mathrm{Light}}_{R}$ and $S{\text{-}\mathrm{Light}}_{C}$ dataset, we assumed that the glass was made of transparent material.

Kim et al. [47] obtained depth images of the image behind the glass and the image reflected on the glass. They proposed a technique that maps the original image onto an object file converted from the depth image to a 3D mesh file for rendering. In this study, the images behind the glass were used for the proposed dataset, and ZoeDepth [55] was used to render the dataset for obtaining depth maps. Fan et al. [38] proposed a technique that applies a Gaussian filter to the reflection data because the reflection images are blurry. To apply the technique proposed by Fan et al., we generated various reflection images based on the transmission images acquired from Kim et al. Building on these two methods, we constructed an image set consisting of 5754 pairs, including $I_{t}$, $I_{td}$, and $I_{r}$ images. In conclusion, the S-light DB is composed of three categories: $S{\text{-}\mathrm{Light}}_{R}$, $S{\text{-}\mathrm{Light}}_{S}$, and $S{\text{-}\mathrm{Light}}_{C}$, with each dataset containing 5754, 3595, and 5794 pairs, respectively.

### 3.2. Multi-Scale NCC

Many researchers have traditionally used performance metrics like the peak signal-to-noise ratio (PSNR) and the structural similarity index (SSIM) that consider the similarity with reference images to compare the performance of specular highlight and reflection removal deep learning models. Due to the limitation of not being able to acquire diffuse images in real-world settings, there is a problem with using PSNR and SSIM as performance metrics for the technique. This issue is the biggest problem when considering integrated specular removal.

The problem addressed in this study was separating or removing integrated specular images from the intensity image. Therefore, performance metrics that consider Cases 2 and 3 are needed. We paid attention to the relationship of the components considered in the two removal tasks. In specular highlight removal, diffuse and specular components occurring on the object surface are independent, and the independence of diffuse and specular components has been widely applied in non-learning methods [14,17]. For this reason, it can generally be said that the correlation between diffuse and specular components is low. In reflection removal, the transmission and reflection components are considered to originate from different scenes, which generally results in a low correlation between them. Techniques that leverage this characteristic exist. Therefore, we aimed to quantitatively evaluate the performance of removal techniques by considering the correlation between the diffuse component or the transmitted diffuse component and the specular reflection component in Cases 2 and 3.

Due to the limitations of using similarity metrics such as PSNR and SSIM, which require a reference image, we cannot apply it to real images. Thus, while traditional metrics are useful for their simplicity and general applicability, MS-NCC’s independence from reference images make it a more effective measure for evaluating specular reflection removal techniques. Furthermore, general correlation techniques typically consider only the overall correlation between variables. Such correlations cannot account for the local region correlations between specular and diffuse components that occur in opaque objects. The spatial information reflected in MS-NCC addresses this issue and has the advantage of allowing visualization of the correlation between the two components. Figure 9 provides example images showing correlation maps based on the window size, depicting the results of applying MS-NCC to the output and residual images of the deep learning model.

Equation (Equation 5) expresses the average map, $M{\left(i,j\right)}_{f}$, of pixels within a window of odd size (k×k) in the image *f*, where (i,j) denotes the pixel coordinates, and the ranges for *u* and *v* are from −(k−1)/2 to (k−1)/2.
(5)$$\begin{array}{c} M{\left(i,j\right)}_{f}=\frac{1}{k^{2}}\sum_{u}\sum_{v}f\left(i+u,j+v\right) \end{array}$$
Equation (Equation 6) presents a formula that indicates the variability in the input image. When the input images are the same, it represents the local variance, whereas it corresponds to the local covariance when the input images are different.
(6)$$\begin{array}{c} V{\left(i,j\right)}_{f}=\frac{1}{k^{2}}\sum_{u}\sum_{v}f\left(i+u,j+v\right)\cdot g\left(i+u,j+v\right)-M{\left(i,j\right)}_{k}\cdot M{\left(i,j\right)}_{g} \end{array}$$
Equation (Equation 7) represents the local normalized cross-correlation (Local-NCC) between the input images, *f* and *g*, based on their local information.
(7)$$\begin{array}{c} L\text{-}NCC{\left(k\right)}_{f,g}=\left|mean\left(\frac{V{\left(i,j\right)}_{f,g}}{\sqrt{V{\left(i,j\right)}_{f,f}\cdot V{\left(i,j\right)}_{g,g}}}\right)\right| \end{array}$$
The window size, denoted as $k_{n}$, is determined based on the smaller dimension (either height or width) of the input image, and this $k_{n}$ window size will be the same size as $2^{\left(-n+1\right)}\times 2^{\left(-n+1\right)}$ of the smaller dimension. Multi scale-NCC was calculated as the weighted sum of local-NCC:(8)$$\begin{array}{c} MS\text{-}NCC=\sum_{n=1}^{5}\lambda_{n}\cdot L\text{-}NCC{\left(k_{n}\right)}_{f,g} \end{array}$$
In Equation (Equation 8), the weight value, $\lambda_{n}$ has been uniformly set. The MS-NCC equation mainly considers the global independence of the transmission and reflection images and the local independence between the diffuse and specular components within objects.

The range of correlation lies between 0 and 1. A range of less than 0.2 from 0 indicates a very weak or no correlation, from 0.2 to less than 0.4 indicates a weak correlation, and from 0.4 to less than 0.6 indicates a moderate correlation. Ranges beyond this signify a very high correlation. Therefore, an MS-NCC output value of less than 0.2 signifies a successful separation. Figure 10 visualizes the difference between varying correlation values according to the above mentioned threshold ranges. The bottom images of D and S (residual between input and model output) are input of MS-NCC. Although the RGB image of residual seems dark, it does not mean that the pixel value of the residual image is absolute zero (See S from Figure 10).

## 4. Experiments

There are no reports on a single-image-based deep learning network that simultaneously addresses the removal of specular images on object surfaces and images containing reflections on glass, including transmitted specular components. Therefore, experiments were conducted using models proposed for the existing problems of specular highlight removal and reflection removal. We have followed the same training strategy and hyperparameters as mentioned in the main papers of the models we have used. The models used in this experiment (also discussed in Section 2) are as follows:-**Specular Highlight Removal Network:** [35,36,37]-**Reflection Removal Network:** [41,42,44,45]

### 4.1. Training and Evaluation in Specular Highlight Removal Network

This experiment assessed the generalizability of specular highlight removal networks and their suitability for training on the dataset constructed in this study. The common characteristics of the specular highlight removal networks used in the experiment include detecting the removal region and removing the reflection component based on the detected area. Mask images indicating specular locations are required to train detection-based deep learning models. This study did not construct mask images separately for the Case 2 dataset. Therefore, the technique proposed in the SHIQ [34] paper was applied to produce binary mask images for this dataset. The training and test datasets were split at an 80:20 ratio for the experiment.

Table 2 and Table 3 present the performance of models trained on each dataset when tested on their respective test datasets. This experiment adopted PSNR, SSIM, NCC, and MS-NCC as the performance metrics. The output image $I_{d^{\prime }}$ and the specular image $I_{s^{\prime }}\left(=I-I_{d^{\prime }}\right)$ of the model were used to compute NCC and MS- NCC; both images were converted to grayscale images.

In general, the detection-based specular highlight removal networks used in this experiment showed higher performance on the previously proposed datasets, SHIQ [34] and PSD [35], than the dataset constructed in this paper. This trend aligns with expectations. Initially, the SHIQ [34] and PSD [35] datasets were constructed assuming that specular images were composed mostly of white highlights or produced in controlled experimental environments using single lighting conditions. As a result, the existing SHIQ [34] and PSD [35] datasets share common features that can aid deep learning models in specular region detection training.

On the other hand, the dataset constructed in this study consisted of various lighting conditions and objects, resulting in a wide range of specular components. Experiments confirmed that specular region detection training was not as effective on the proposed dataset. Figure 11 provides examples of models trained with the proposed dataset during testing.

### 4.2. Training and Evaluation in Reflection Removal Network

This experiment assessed the generalizability of reflection removal networks and their suitability for training with the dataset constructed in this study. Table 3 summarizes the performance of models trained on each dataset when tested on their respective test datasets, following a similar structure to that in Table 2. The models trained with the current dataset exhibited better performance across PSNR, SSIM, NCC, and MS-NCC metrics than previously conducted specular removal models.

Furthermore, the NCC and MS-NCC values for the SHIQ [34] and PSD [35] test datasets either outperformed or closely matched those trained on the same dataset. Figure 12 presents example images from the model trained on the present dataset when applied to SHIQ [34] test images. Several network output images exhibited a more natural appearance and effectively removed specular components occurring on smooth materials, such as glass and metal, compared to the ground truth images.

### 4.3. Training on the Constructed Dataset and Evaluation on Real Images

This experiment assessed the “trends” in whether the deep learning models trained on the proposed dataset show significant performance in real-world scenarios. An experimental model designed for reflection removal was used to train the dataset proposed in this paper, which considers reflection and surface specular components. Several reasons support this choice. First, experiment 4.1 showed that a dataset focused solely on surface specular components is not suitable for training a specular highlight removal model. As illustrated in Figure 8, unlike SHIQ and PSD, $S{\text{-}\mathrm{Light}}_{S}$’s features areas of various sizes and distribution of colors are unsuitable for specular highlight removal models that are designed to focus on detecting specular regions, leading to ineffective learning, as evidenced by Table 2. Second, according to the observation from experiment 4.2, the reflection removal models also showed versatility when applied to the SHIQ [34] and PSD [35] test datasets. Unlike specular removal methods, reflection removal models mainly consider global features of the images. Thus, the mentioned rationale for experiment 4.1 also confirms the findings from experiment 4.2. Third, the reflection components constructed in this paper are present throughout the entire image, causing detection-based specular removal networks to be unsuitable. Three distinct settings were used in the training methodology for evaluating specular highlight removal and reflection removal tasks:1.**Half Setting**: In the first setting, 50% of the data were randomly selected from the $S{\text{-}\mathrm{Light}}_{R}$ and $S{\text{-}\mathrm{Light}}_{C}$ datasets.2.**Combined Setting**: The second setting exclusively used the $S{\text{-}\mathrm{Light}}_{C}$ dataset.3.**Pretrained Setting**: The third setting used off-the-shelf models, provided by the authors to evaluate the specular and reflection removal tasks.

In each setting, well-known reflection removal benchmarks were used to evaluate the models on real-world images in Cases 2 and 3: the SIR-Wild and Solid object datasets. The SIR dataset consisted of intensity, transmission, and reflection images. The transmission and intensity images were used for Cases 2 and 3, respectively. NCC and MS-NCC were used as evaluation metrics to compare the correlation between the model output image and the residual image because there are no reference images in the considered scenarios, similar to experiments 4.1 and 4.2.

Table 4 presents the quantitative evaluations of the Wild and Solid Object datasets. Overall, the correlation between the model output image and the residual image was lowest for models trained in the “Half setting”. However, on the other side, the models trained under “Combined setting” performed decently good compared to the “Pretrained setting” where it was more biased towards the reflection removal task.

Similarly, Figure 13 illustrates the outputs from each setting trained on the model proposed by Li et al. [45]. The figure illustrates that the model, when trained in a “Half setting” condition, effectively eliminates reflection and specular highlight components from the input images. Conversely, in the “Pretrained setting” condition, it struggles to fully remove specular highlights. This challenge arises because the datasets employed for the pretrained models are predominantly aligned with reflection removal, leading to difficulties in generalizing the task of specular highlight elimination. In the “Combined setting”, the model achieves moderate performance on the provided input images.

To assess the generalization performance of the proposed DB in the context of specular highlight removal, we conducted experiments using reflection removal models trained under the “Half setting” condition, which had previously demonstrated superior performance. The PSD dataset, acquired through a polarizing camera, was employed as the evaluation dataset. Given the availability of precise GT images, we utilized PSNR and SSIM. Additionally, we considered MS-NCC and NCC as evaluation metrics. Furthermore, we applied a pretrained model, previously utilized, to the PSD. Table 5 presents the performance across each metric. A notable result is that models trained under the “Half setting” condition show superior results in MS-NCC and NCC, but the pretrained model performs better in terms of PSNR and SSIM. We visually confirmed that the output images from the pretrained model showed no significant differences from the input images.

Consequently, we also measured the PSNR and SSIM values with the input image as a reference, the results of which are enclosed in parentheses in Table 5. Overall, PSNR recorded approximately 30, and SSIM about 0.97. These results imply a minimal difference in pixel values between the intensity images and GT images (diffuse component), directly relating to our previous mention (in Section 3.1.1) that the dataset primarily consists of specular components with narrow area sizes. Additionally, the models trained under the “Half setting” generally recorded lower scores in traditional performance metrics, suggesting that training solely on the proposed DB might be limited in removing local specular reflection components. As reflection removal models have tendency to consider an entire region, they lack the ability to learn features from the local regions of the images. Figure 14 visually presents the output results of the half setting model in Cases 2 and 3.

## 5. Conclusions

Specular reflection removal was categorized into three main scenarios, and a dataset tailored for training single-image-based deep learning models was proposed to address these scenarios. Experiments that were divided into two categories were conducted to investigate suitable networks for the proposed *S*-Light dataset: one using an area detector-based specular highlight deep learning model and the other using a reflection removal deep learning model. The reflection removal model trained on the *S*-Light*_s_* dataset exhibited generalizability, as confirmed by the previously proposed metrics, such as SHIQ and PSD. Furthermore, a quantitative measure called MS-NCC, which relies on the correlation between the diffuse and specular components, was used to evaluate the network’s performance. This measure was effective when applied to real-world images from datasets, such as the $SIR^{2}$ Wildscene and Solid object. Therefore, for the effective removal of specular reflections, it is essential to construct an appropriate dataset.

In conclusion, three major research areas necessary for comprehensive specular removal were addressed. First, the establishment of training datasets was achieved. Second, the research into deep learning models was suitable for integrated specular removal. Finally, a correlation metric that utilizes the relationship between specular and diffuse components was developed for more detailed quantitative evaluation in real-world scenarios. However, in our present work, we consider the dataset’s size as a limitation due to the need of a larger dataset considering various environmental conditions that can help the deep learning models to be more robust towards the real life images and increase the generalization ability. Future studies will also focus on proposing novel deep learning models for unified specular reflection removal, as the present deep learning networks are not able to tackle the local region information, causing potential non-efficient removal.

## Figures and Tables

**Figure 1 sensors-24-02286-f001:**
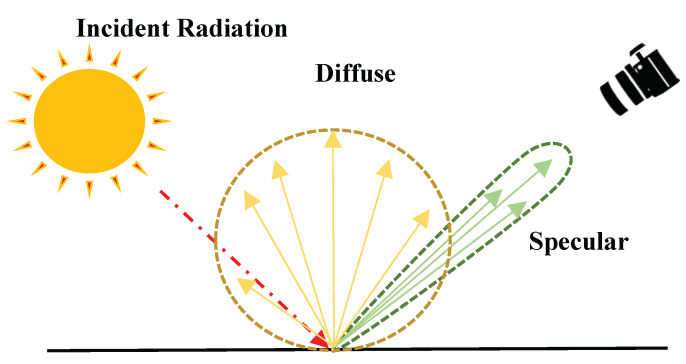
Diffuse and specular reflection occurring on the surface of an object.

**Figure 2 sensors-24-02286-f002:**
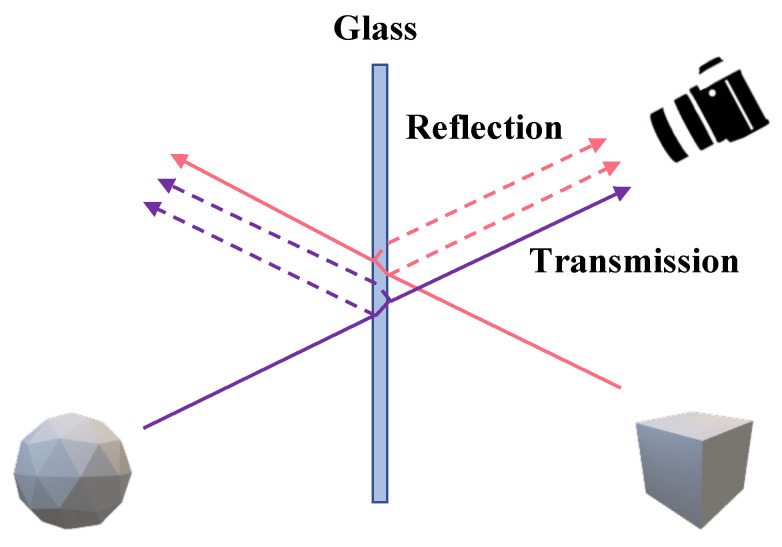
Diffuse and glass reflection.

**Figure 3 sensors-24-02286-f003:**
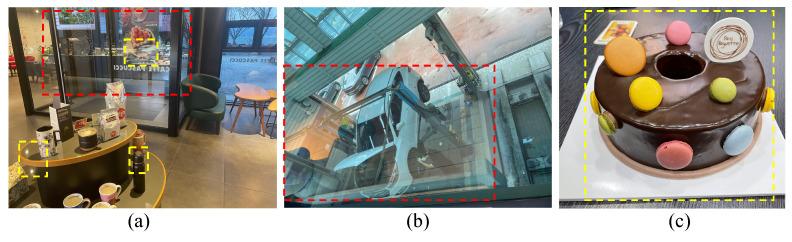
Example images with specular reflection. The red box marks specular reflection on a transparent material, and the yellow box shows it on an opaque material: (**a**) case occurring in transparent and opaque materials, (**b**) case occurring in transparent materials, and (**c**) case occurring in opaque materials.

**Figure 4 sensors-24-02286-f004:**
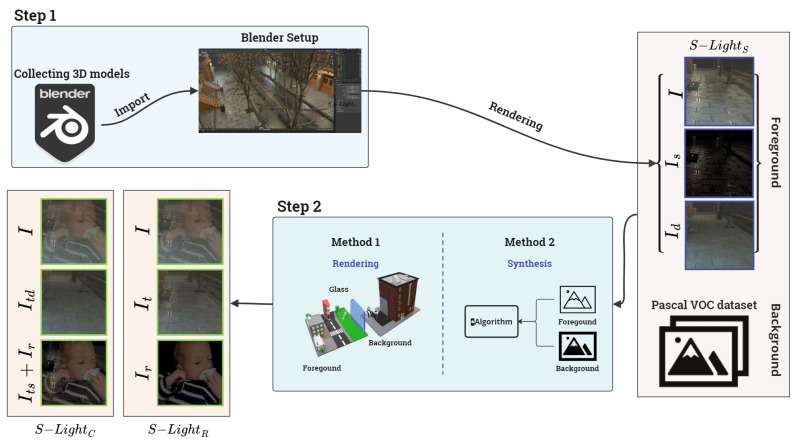
Overall workflow of *S*-Light DB construction. After each step, $S\text{-}{Light}_{S}$ and $S\text{-}{Light}_{R\&C}$ datasets were produced. In the figure, we have used [47] for ’Method 1’ and [38] for ’Method 2’.

**Figure 5 sensors-24-02286-f005:**
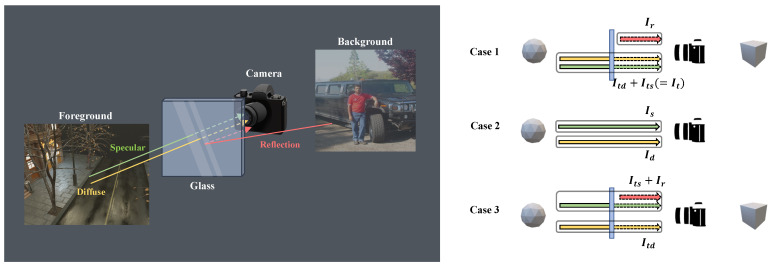
(**Left**) Example image illustrating the captured reflection component from the camera. (**Right**) Cases 1 and 2 briefly represent the problems that reflection removal and specular highlight removal aim to address. Case 3 is an extension of Case 1, where the target is the diffuse component.

**Figure 6 sensors-24-02286-f006:**
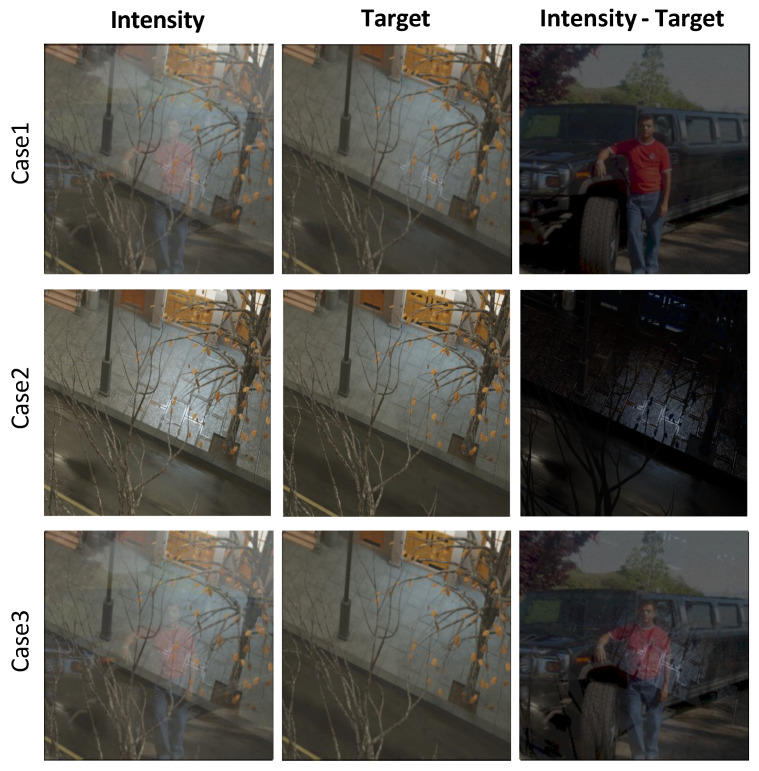
Example images from the proposed dataset.

**Figure 7 sensors-24-02286-f007:**
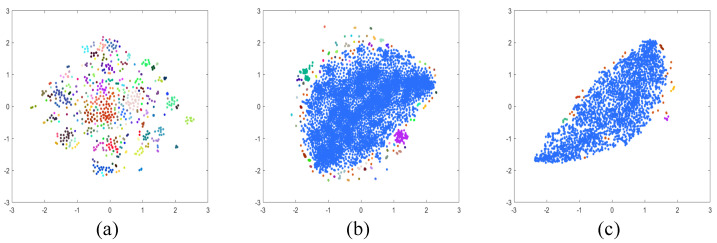
Results of applying t−SNE and DBSCAN clustering: (**a**) PSD dataset, (**b**) SHIQ dataset, and (**c**) $S{\text{-}\mathrm{Light}}_{S}$ dataset. The each color refers to the cluster ID from DBSCAN algorithm.

**Figure 8 sensors-24-02286-f008:**
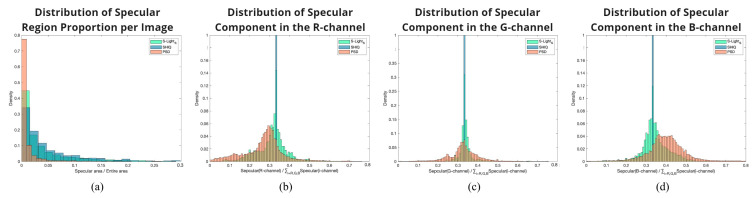
Visualizing the distribution of the specular component: (**a**) proportion of specular regions per image, and (**b**–**d**) specular intensity across RGB channels, utilizing probability density functions (PDF).

**Figure 9 sensors-24-02286-f009:**
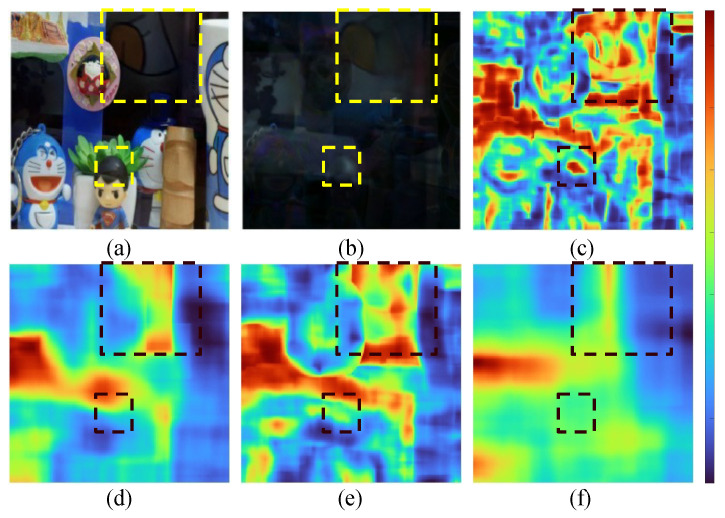
In the z-direction, (**a**) model output image, (**b**) residual image, and (**c**–**f**) correlation map after applying the small ($k_{5}$) to large window ($k_{2}$) respectively. Please refer to the $L\text{-}NCC{\left(k_{n}\right)}_{f,g}$ in Equation (Equation 8).

**Figure 10 sensors-24-02286-f010:**
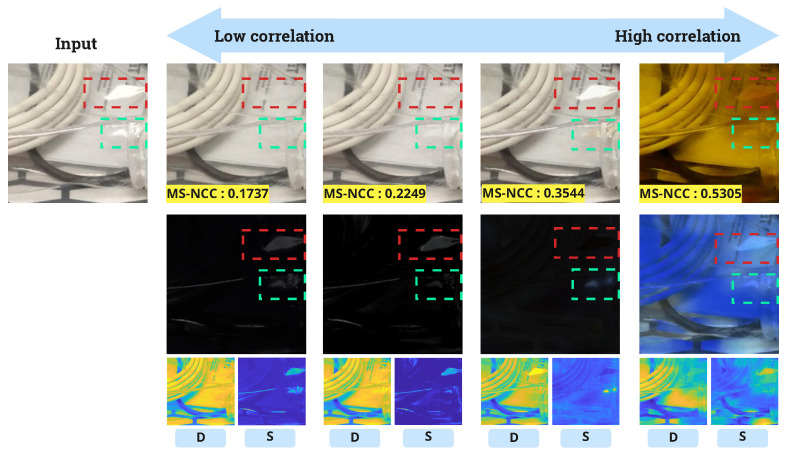
Comparative visualization of output images across correlation value ranges (**Right**: high correlation and **Left**: low correlation). The bottom pair of images are grayscale of diffuse (**D**) and specular (**S**).The bounding boxes (red and green) indicate specular reflection region.

**Figure 11 sensors-24-02286-f011:**
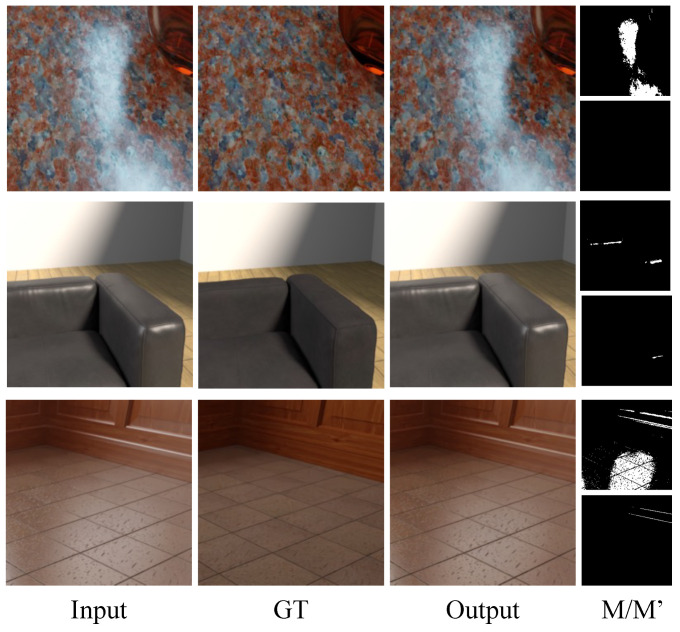
Example images from the region detection-based model [35] M/M’ represents mask GT (ground truth) and mask output in each row.

**Figure 12 sensors-24-02286-f012:**
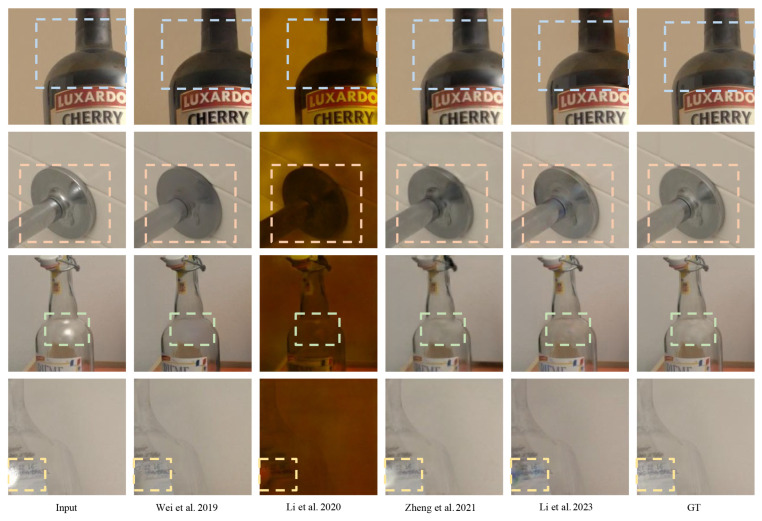
Examples of specular removal results [41,42,44,45] on the SHIQ test set after training the dataset. (GT: ground truth).Bounding boxes mean the region of specular reflection.

**Figure 13 sensors-24-02286-f013:**
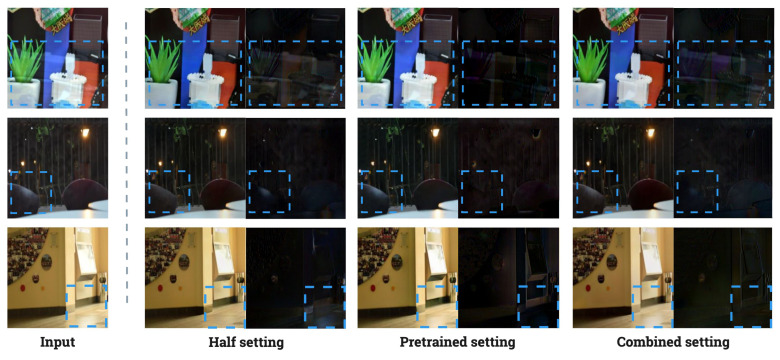
Visualization of example outputs from each setting: half, pretrained and combined. Bounding boxes depict the regions of specular reflection.

**Figure 14 sensors-24-02286-f014:**
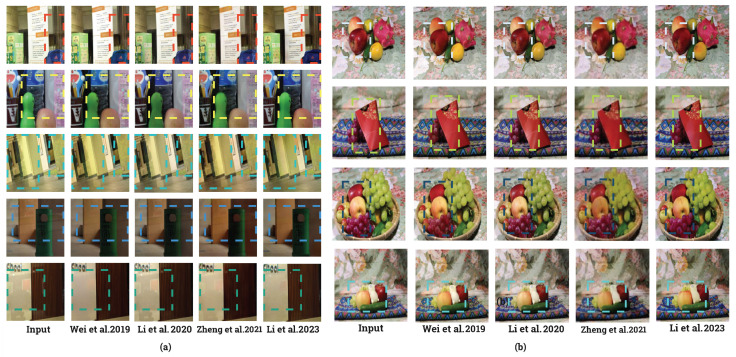
Half setting model results: (**a**) input and output images [41,42,44,45] of Case 3, (**b**) input and output images [41,42,44,45] of Case 2.

**Table 1 sensors-24-02286-t001:** Seven hundred and nineteen scenes were generated in rendering scenes where the objects and backgrounds do not overlap. (I, S, D, and M stand for Intensity, Specular, Diffuse, and Mask).

	Number of Images	Image Set	Specular Color	Clustering Portion	Scene	Acquisition
PSD [35]	13,380	I, D	Multiple	0.007	2210	Captured
SHIQ [34]	10,825	I, D, S, M	Single	0.9	1016	Captured + RPCA [3]
*S*-Light*_s_*	3595	I, D	Multiple	0.96	719	Rendering

**Table 2 sensors-24-02286-t002:** Evaluation of specular highlight removal models trained on each dataset when assessed on the corresponding test datasets. The training dataset is indicated next to the model names on the right. The test dataset is listed at the top. When the training and test datasets differ, they are highlighted in green. “Mean” represents the average value within the highlighted green area. The best-performing values for each model/test dataset combination are shown in bold.

		*S*-Light*_s_* (Proposed DB)				SHIQ [34]				PSD [35]				Mean			
		PSNR↑	SSIM↑	NCC↓	MS-NCC↓	PSNR↑	SSIM↑	NCC↓	MS-NCC↓	PSNR↑	SSIM↑	NCC↓	MS-NCC↓	PSNR↑	SSIM↑	NCC↓	MS-NCC↓
Wu [37]	*S*-Light*_s_*	15.895	0.666	0.614	0.468	19.954	0.861	0.643	0.728	17.417	0.754	0.731	0.872	18.685	0.807	0.687	0.800
	SHIQ [34]	22.657	0.785	0.512	0.498	**30.754**	**0.950**	0.285	0.581	23.931	0.888	0.446	0.516	23.294	0.837	0.479	0.507
	PSD [35]	**24.143**	**0.844**	**0.235**	**0.324**	24.374	0.908	**0.249**	**0.231**	**29.049**	**0.951**	**0.201**	**0.215**	**24.258**	**0.876**	**0.242**	**0.277**
Wu [35]	*S*-Light*_s_*	25.372	**0.868**	0.309	0.419	25.817	0.927	0.321	0.459	28.113	0.946	0.111	0.110	**26.965**	**0.936**	0.216	0.284
	SHIQ [34]	24.590	0.844	0.310	0.271	**32.551**	**0.961**	**0.180**	0.292	25.919	0.930	0.256	0.262	25.254	0.887	0.283	0.266
	PSD [35]	**25.870**	0.862	**0.150**	**0.166**	26.971	0.929	0.175	**0.163**	**30.155**	**0.958**	**0.137**	**0.130**	26.420	0.895	**0.162**	**0.165**
Hu [36]	*S*-Light*_s_*	**24.575**	**0.825**	**0.196**	**0.285**	22.732	0.863	0.407	0.454	25.398	0.892	0.189	0.229	**24.065**	**0.877**	0.298	**0.342**
	SHIQ [34]	21.475	0.776	0.398	0.517	**27.286**	**0.944**	**0.192**	**0.345**	24.360	**0.901**	**0.176**	**0.208**	22.917	0.839	**0.287**	0.362
	PSD [35]	21.102	0.794	0.351	0.499	21.09	0.862	0.435	0.719	**24.631**	0.890	0.187	0.267	21.09	0.828	0.393	0.609

**Table 3 sensors-24-02286-t003:** Evaluation of reflection removal models trained on each dataset when assessed on the corresponding test datasets. The training dataset is indicated next to the model names on the right. This table is identical in structure to Table 2.

		*S*-Light*_s_* (Proposed DB)				SHIQ [34]				PSD [35]				Mean			
		PSNR↑	SSIM↑	NCC↓	MS-NCC↓	PSNR↑	SSIM↑	NCC↓	MS-NCC↓	PSNR↑	SSIM↑	NCC↓	MS-NCC↓	PSNR↑	SSIM↑	NCC↓	MS-NCC↓
Wei [44]	*S*-Light*_s_*	**27.869**	**0.898**	**0.184**	**0.225**	23.621	0.890	**0.224**	**0.248**	26.030	0.921	**0.100**	0.203	24.826	0.906	**0.162**	**0.225**
	SHIQ [34]	23.518	0.810	0.569	0.372	**32.217**	**0.960**	0.478	0.452	22.809	0.886	0.497	0.324	23.164	0.848	0.533	0.348
	PSD [35]	24.824	0.850	0.211	0.283	26.011	0.918	0.241	0.284	**29.815**	**0.957**	0.178	**0.147**	**25.418**	**0.884**	0.226	0.284
Li [41]	*S*-Light*_s_*	16.602	0.528	0.388	0.464	12.685	0.490	0.535	**0.350**	13.844	0.560	0.504	0.520	13.265	0.525	0.520	0.435
	SHIQ [34]	**24.761**	0.844	0.209	0.452	**32.634**	**0.964**	**0.200**	0.418	26.500	0.928	0.248	0.533	25.630	**0.886**	0.229	0.492
	PSD [35]	24.351	**0.853**	**0.170**	**0.405**	24.297	0.919	0.283	0.382	**27.593**	**0.949**	**0.083**	**0.322**	**24.324**	0.885	**0.226**	**0.394**
Zheng [42]	*S*-Light*_s_*	**27.162**	**0.867**	**0.266**	**0.247**	25.054	0.890	0.175	**0.249**	24.619	0.886	**0.118**	0.213	**24.837**	**0.888**	**0.147**	**0.231**
	SHIQ [34]	24.192	0.805	0.368	0.372	**35.309**	**0.965**	**0.141**	0.342	**24.729**	0.897	0.334	0.386	24.460	0.851	0.351	0.379
	PSD [35]	18.177	0.719	0.355	0.344	18.586	0.764	0.326	0.511	21.762	**0.918**	0.197	**0.172**	18.382	0.742	0.340	0.427
Li [45]	*S*-Light*_s_*	**25.767**	**0.850**	0.241	0.229	25.354	0.926	**0.170**	**0.213**	24.907	0.926	**0.133**	**0.178**	**25.130**	**0.926**	**0.151**	**0.195**
	SHIQ [34]	22.491	0.797	0.231	0.173	**30.491**	**0.958**	0.210	0.234	22.440	0.900	0.175	0.294	22.465	0.848	0.203	0.234
	PSD [35]	23.787	0.833	**0.222**	**0.227**	23.386	0.910	0.222	0.309	**28.872**	**0.957**	0.131	0.188	23.586	0.872	0.222	0.268

**Table 4 sensors-24-02286-t004:** Validation results in $SIR^{2}$ Wildscene and $SIR^{2}$ Solid object. M and T represent the Mix and Transmission image sets. Results with better performance in the half, combined and pretrained settings are indicated in bold.

Model		Wildscene M [50]		Wildscene T [50]		Solid Object M [50]		Solid Object T [50]	
		NCC ↓	MS-NCC ↓	NCC ↓	MS-NCC ↓	NCC ↓	MS-NCC ↓	NCC ↓	MS-NCC ↓
Wei [44]	Half	**0.1437**	0.2496	0.1547	0.2129	**0.1181**	**0.1712**	**0.1588**	**0.1375**
	Combined	0.1618	**0.2153**	**0.1474**	**0.2029**	0.2638	0.2744	0.2472	0.3444
	Pretrained	0.1709	0.2991	0.2947	0.3829	0.1886	0.2496	0.3317	0.2915
Li [41]	Half	0.1301	0.2376	0.1469	0.2451	**0.1271**	**0.1610**	**0.1555**	**0.1254**
	Combined	**0.1277**	**0.1738**	**0.1225**	**0.1659**	0.1448	0.1668	0.1991	0.2100
	Pretrained	0.3753	0.1886	0.4261	0.2051	0.4715	0.1937	0.5167	0.2231
Zheng [42]	Half	**0.1284**	**0.2019**	**0.1382**	**0.2023**	**0.1161**	**0.1267**	**0.1229**	**0.1395**
	Combined	0.1441	0.2589	0.1433	0.2588	0.1497	0.2650	0.1649	0.3547
	Pretrained	0.1505	0.2525	0.1824	0.3139	0.4294	0.3829	0.2893	0.3311
Li [45]	Half	0.1667	**0.1549**	0.1689	**0.1786**	0.1376	**0.1448**	**0.0842**	**0.1354**
	Combined	**0.1484**	0.2059	**0.1438**	0.2578	**0.1071**	0.2082	0.1229	0.1467
	Pretrained	0.1633	0.2951	0.2097	0.2459	0.2448	0.2588	0.3159	0.3054

**Table 5 sensors-24-02286-t005:** Validation results of PSD under half and pretrained settings. The values in parentheses represent the PSNR and SSIM when the input image is used as the reference.

Model		PSD [35]			
		PSNR ↑	SSIM ↑	NCC ↓	MS-NCC ↓
Wei [44]	Half	25.5000	0.9155	0.1037	0.2350
	Pretrained	27.9452 (29.8697)	0.9452 (0.9740)	0.2054	0.3957
Li [41]	Half	26.2059	0.9200	0.1520	0.2600
	Pretrained	25.6764 (26.6873)	0.9423 (0.9619)	0.3828	0.3459
Zheng [42]	Half	22.3527	0.8673	0.1296	0.2546
	Pretrained	26.4305 (30.6514)	0.9407 (0.9760)	0.2729	0.3356
Li [45]	Half	22.1738	0.9073	0.1395	0.2395
	Pretrained	26.6873 (30.1141)	0.9389 (0.9695)	0.4707	0.5057

## Data Availability

The data presented in this study are available in https://drive.google.com/file/d/1Ntjq4KTprVVum8ykYp3HQ9PNVGHijZ4v/view?usp=sharing (accessed on 18 March 2024).

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
