# Peer review of "S-LIGHT: Synthetic Dataset for the Separation of Diffuse and Specular Reflection Images"

_sensors, 2024, doi:10.3390/s24072286_

Round 1

Reviewer 1 Report

Comments and Suggestions for Authors

In this paper, two related but independent studies on specular highlight removal and reflection removal in computer vision are integrated, and a new task is proposed to solve the problem of coexistence of these two types of specular effects. To support the further research of this new task, a synthetic dataset for deep learning model training is constructed. And a measure index of model learning effect is designed.

Improvement suggestions:

1. The premise of the new task of integrating the two studies of image reflection removal and specular highlight removal discussed in this paper is that the reflection problem and specular highlight problem will coexist in the image, but the introduction lacks corresponding information to support this premise, so it is suggested to add relevant content to make significance of this paper more convincing. For example, it is explained in which specific imaging scenes the problem of reflection and specular highlight will occur at the same time, and the corresponding image sample is inserted into the text;

2. The explanation of principles and related work in sections 1.1. and 1.2. is too redundant and needs to be further refined;

3. The statement in the first paragraph of section 2.1. is unclear and may be incorrect, and does not correspond to the following text and the figure;

“In each task, the target consisted of the transmission image  and the diffuse image .”

4. The differences between the constructed Case1 subdataset and the Case3 subdataset need to be further clarified in Section 2.1.;

5. Is the dataset diversity evaluation method based on the reduction and Clustering algorithm and the indicator "Clustering portion" in Section 2.1.1. an innovation of this paper? If so, it is necessary to further explain the rationality of this evaluation method; if not, it is necessary to give the source of this evaluation method by quoting references;

6. The description of the second step of the data set construction process in Section 2.1.2. is too sketchy, and it is suggested to visualize the specific process by drawing schematic diagrams;

7. In the first paragraph of section 2.2, the explanation of the defects of using PSNR and SSIM as performance evaluation indexes of specular reflection removal models is not clear enough;

8. The content of the experiment in section 3.2 is incorrect. SHIQ data set and PSD dataset are applied to the specular highlight removal task, but the experiments in Section 3.2 focus on the reflection removal task. so SHIQ dataset and PSD dataset should not be used here, but the dataset corresponding to the reflection removal task should be used?

9. The experiment in section 3.3 can be improved,

â‘  This section only completed the model generalization performance evaluation experiment that applied the constructed dataset to solve the reflection removal problem, but did not complete the model generalization performance evaluation experiment that applied the constructed dataset to solve the specular highlight removal problem;

â‘¡ In this section, only four reflection removal algorithms were successively trained on the data set constructed in this paper and tested on the SIR-Wild and Solid object datasets. It is not proved that the reflection removal algorithm trained on the dataset constructed in this paper has stronger generalization performance than the same algorithm trained on the dataset related to other reflection removal tasks. It is suggested that the same reflection removal algorithm should be trained by using the dataset constructed in this paper and other datasets related to reflection removal problems, and then the same test set should be used for testing, calculation and comparison of evaluation indicators, so as to verify that the dataset constructed in this paper is superior to other datasets related to reflection removal tasks.

 Author Response

Dear Reviewer, 

Thank you so much for taking time amidst your busy schedule. I appreciate your comments and feedback on our manuscript. I have corrected your mentioned point and modified the manuscript accordingly.

Thank you again for your valuable input on our manuscript which helped us enhance the quality of the paper.

Sincerely, Sangho Jo

Reviewer 2 Report

Comments and Suggestions for Authors

The paper proposes a synthetic dataset called S-Light targeting integrated specular reflection removal using single image deep learning models. A novel performance metric called Multi-Scale Normalized Cross-Correlation (MS-NCC) is also introduced to evaluate specular separation quantitatively.

The S-Light dataset is rendered using 3D models under diverse illuminations and contains surface specular highlights, reflections, and combinations of both. Experiments show reflection removal models trained on S-Light exhibit generalization compared to existing datasets like SHIQ and PSD.

Real image evaluation uses the MS-NCC metric to measure correlation between separated diffuse and specular components. Models trained on S-Light maintain relatively lower correlation compared to half-dataset variants, demonstrating its effectiveness.

While promising, some issues need further investigation - analyzing model generalization differences, output quality assessment, limitations of synthetic data, sensitivity of the MS-NCC metric, proportions of reflection vs surface specular images.

The concept shows potential in unifying the field of specular removal. As future work, specialized deep network architectures can leverage the diversity in S-Light to improve real-world performance. Addressing limitations and constructing larger datasets would also be impactful.

The core ideas around integrated removal, diverse synthetic data and correlation-based evaluation are valuable. With refinements in analysis and scope, this could make notable research contributions. The following comment can enhance the overall quality of the paper and it can be helpful to the readers:

  1. The literature review needs to be expanded to include more recent works in integrated specular reflection removal using deep learning techniques.
  2. Additional comparisons with state-of-the-art methods on benchmark datasets would strengthen the evaluation.
  3. More analysis is needed on the limitations of existing datasets and how the proposed dataset aims to address those limitations.
  4. Expand the explanation for how the synthetic dataset renders various lighting conditions and object types to induce diverse specular effects.
  5. Include quantitative metrics or visualizations that demonstrate the diversity of specular effects in the proposed dataset.
  6. Clarify how the models used in the experiments were trained, optimized, and evaluated. Details like architecture, training procedures, loss functions are needed.
  7. The generalizability experiment results need more analysis and discussion on why performance differed between datasets.
  8. Provide examples of model outputs to illustrate cases where the proposed dataset enabled improved generalization.
  9. Explain the rationale for choosing the reflection removal model architecture used for real image experiments.
  10. Analyze and discuss possible reasons for the performance difference between Half and Combined experiments on real images.
  11. Visualize more output images from real image experiments to subjectively assess specular removal quality.
  12. Expand the analysis on correlation metrics as evaluation criteria and their effectiveness compared to similarity metrics.
  13. Clarify which specific components are considered for correlation computation in the MS-NCC formulation.
  14. Discuss challenges in acquiring ground truth diffuse images for real images and alternatives for quantitative evaluation.
  15. Compare the MS-NCC metric against other correlation measures in terms of efficacy for this application.
  16. Examine the sensitivity of MS-NCC threshold values for determining successful separation of specular and diffuse components.
  17. Provide more details on the types, numbers and proportions of images with reflection vs. surface specular effects in the dataset.
  18. Discuss potential limitations of the proposed approach and dataset.
  19. Highlight the most significant open problems that need to be addressed in future works on unified specular reflection removal.
  20. Proofread the paper thoroughly to fix typos, formatting, grammar issues, and improve clarity.

Author Response

Dear Reviewer, 

Thank you so much for taking time amidst your busy schedule and give such detailed feedbacks and suggestions. I appreciate your comments  on our manuscript. I have corrected your mentioned point and modified the manuscript accordingly.

Thank you again for your valuable input on our manuscript which helped us enhance the quality of the paper.

Sincerely, Sangho Jo

Reviewer 3 Report

Comments and Suggestions for Authors

The synthetic dataset can be a significant contribution only if it is used to create new solutions that provide clear advantages compared to the state-of-the-art. In the paper, this work is planned for the future. Also, the authors provide only indirect evidence that their synthetic dataset correctly represents real diffuse and specular reflections. Thus, it is currently difficult to distinguish if the first contribution is significant or just a set of generated images that do not provide any new scientific or even engineering results.

The second contribution claimed by the authors (a performance metric that considers the relationship between the diffuse component and each specular component to measure the separation level quantitatively) is fair, but in my opinion, the evidence of its novelty and usefulness is not enough to solely justify publishing the whole paper in a reputable journal like Sensors.

I admire the approach the authors proposed and think they are on the right path. At the same time, this result should be published as a part of the research paper describing some new solutions created/trained using the proposed dataset and the metric. Also, the reviewed paper can be published after the paper dedicated to the new solutions is published. In this case, the authors can clearly show that their dataset is valid and representative. Unfortunately, before the current moment, it is too early to evaluate their current result.

Author Response

Dear Reviewer, 

Thank you so much for dedicating your valuable time to our manuscript and providing your insights and detailed feedback. We have carefully read your comments and modified the paper to clarify few points you have mentioned.

We have attached the document. In the document, we have highlighted the modifications. 

Thank you again for your time and consideration

Sincerely, Sangho Jo

Round 2

Reviewer 3 Report

Comments and Suggestions for Authors

The authors have significantly revised the manuscript according to my and other reviewers' recommendations. At the same time, I remain of the opinion that it makes sense to publish a synthetic dataset only after some new useful results have been obtained based on it, and not before.  I understand that the dataset is developed before the new deep learning models, but this does not mean that it should be published immediately after development. However, this problem is not related to the quality of the manuscript, but in general to the possibility of publishing this work in the Sensors journal at this stage of the research. From the fact that the editor did not reject the paper at the previous stage, despite my recommendation, I conclude that he considers the publication of such papers acceptable. In this respect, I see no reason to prevent its publication.